# Program evaluation of a wilderness experience for adolescents facing cancer: A time in nature to heal, connect and find strength

E. Anne Lown[1☯], Heather Rose Otto[2☯], Christine Lynn Norton[3☯], Miek C. Jong[4,5☯], Mats Jong [5☯]*

1 Department of Social Behavioral Sciences, University of California San Francisco, San Francisco, California, United States of America, 2 See You at the Summit, Portland, Oregon, United States of America, 3 School of Social Work, Texas State University, Kyle, Texas, United States of America, 4 National Research Center in Complementary and Alternative Medicine (NAFKAM), UiT - The Arctic University of Norway, Tromsø, Norway, 5 Department of Health Sciences, Mid Sweden University, Sundsvall, Sweden

☯ These authors contributed equally to this work.
* mats.jong@miun.se

**Data Availability Statement:** APPLICATION INSTRUCTION FOR USE OF SEE YOU AT THE SUMMIT EVALUATION DATA The investigators are committed to making the best use of the SYATS

## Abstract

### Objective

Despite advances in cancer treatment and increased survival, adolescents in treatment for cancer often suffer from psychosocial distress, negative mood, and chronic health problems. Wilderness therapy is considered a promising program to address psychosocial issues among adolescents with mental or behavioral health issues. There is little research on whether it may benefit adolescents in cancer treatment.

### Methods

This program evaluation in the form of a pilot study uses qualitative and quantitative measures to describe the feasibility, acceptability, safety, and to explore the impact of a nine-day wilderness program among adolescents aged 13–17 in treatment or who recently finished treatment for a cancer. Quantitative tracking documented recruitment, retention, safety, and participant satisfaction. PROMIS measures assessed mental and social health, positive affect, fatigue, pain interference and intensity over three time-points: pre, post, and three-months after the nine-day wilderness experience. Mean differences were compared over time. Qualitative data collection involved participant observation and open-ended interviews.

### Results

Study enrollment goals were met, enrolling eight adolescent participants with 100% participant retention. No serious adverse events were reported and participants described high satisfaction (9.25/10) with the wilderness experience on the final day and at three-months follow-up (9.5/10). Exploratory data analysis showed scores in a favorable direction

study data while also protecting the confidentiality of the participants. Data will be maintained by the SYATS program however, qualified researchers may access the data. The purpose of this Data Sharing Plan is to ensure that individuals who are interested in using data from the SYATS evaluation will have access to the data and code book, but also so that the use of the data is coordinated, to avoid overlapping analyses and thus wasted time and effort. Researchers interested in accessing the data should submit a letter of intent outlining the proposed project as stipulated below. Approved applicants will receive the codebook to support them in developing a more detailed application to the SYATS Data Review Board. Members of that board include Joshua Hurwitz, Heather Rose Otto, Anne Lown, and Mats Jong. The application for secondary analysis proposals should include the following materials: - Application Form (includes name, position, contact information, project title, CV) - Letter of Intent, including the following: - Rationale - Specific aims - Research design (analysis plan) - Implications for the field - Role of current Investigators (if any) in supporting the proposed study - Sources and amounts of funding to complete the work (existing and proposed) - Timeline for the proposed research (start date, mid-stream progress report, completion date) - Mock tables - Mentoring plan (if applicable) Please direct any questions to the medical director, Dr Joshua Hurwitz and email the completed application to joshua.j.hurwitz@gmail.com A secondary contact is AnneLownHecht@gmail.com The Data Review Board will assess the researcher's capacity to carry out the proposed analyses. Progress on the work will be monitored to ensure that work is completed in a timely fashion and (where there is competing interest) allowing for another investigator to have the opportunity to conduct analyses if the first party does not complete the work according to schedule. Data from observation and qualitative interviews contains sensitive information and is available upon request from the medical director, Dr Joshua Hurwitz joshua.j.hurwitz@gmail.com, and after approval from the Internal Review Board Ethics Committee For Legacy Health (Portland OR, USA).

**Funding:** Funding for (HRO)– the See You at the Summit program was provided in part by the OHSU Knight Cancer Institute Community Partnership Program. (HRO) URL: https://www.ohsu.edu/knight-cancerinstitute/community-partnership-program-grants. Funding for the Evaluation was provided by the University of California San Francisco, Osher Center for Integrative Medicine, Integrative Oncology Research Pilot Award. URL: https://oshercenter.

indicating improved psychosocial outcomes in physical functioning, anxiety, depression, fatigue, and peer relations. From qualitative analysis it is suggested that program participation supported: increased self-confidence and peer connection. The program was evaluated as increasing personal accomplishment, supporting social interaction, having strong staff support, and capitalizing on the natural surroundings.

## Conclusion

Use of a wilderness program is feasible, acceptable, and safe among this highly vulnerable adolescent cancer population. Participants described greater self-confidence and peer connection which developed as participants experienced physical competency, group leadership, and personal strength. Larger randomized controlled studies are needed to learn whether these programs can improve psychosocial outcomes.

## Introduction

### Cancer among adolescents and young adults

Adolescents with cancer have not achieved the same increases in survival as have younger children [1,2], with further disparities in survival rates by race/ethnicity and socioeconomic status (SES) [2,3]. Many factors contribute to this including lower participation in clinical trials including host and disease biology, delayed diagnosis, different treatment approaches, poor compliance and poor adherence to therapy possibly influenced by psychosocial and economic issues [4]. Common cancers impacting adolescents (age 15–19) include brain and nervous system tumors, lymphoma, thyroid, melanoma, leukemia, germ cell cancers, and bone and soft tissue sarcomas [5].

### Needs for adolescent and young adult cancer survivors

Long-term adolescent cancer survivors have greater frailty, lower quality of life, decreased physical capacities, and significant disease and treatment-related health issues in adulthood [6–10]. At least 66% of child and adolescent cancer survivors have one or more chronic diseases as defined by the Common Terminology Criteria for Adverse Events (version 3) [11,12] including conditions such as congestive heart failure, major joint replacement, second malignant neoplasms, severe cognitive dysfunction, coronary artery disease, or renal failure [13].

Adolescents with cancer may be particularly susceptible to social, behavioral, and emotional comorbidities [14] including depression, anxiety, negative body image, chronic health problems, and social isolation [15,16]. It has also been reported that they have a higher risk for the development of stress-related mental health disorders [17], social, academic and vocational difficulties [18,19], secondary cancers [20,21] as well as increased risky health behaviors [22–24]. Sedentary behavior and obesity are higher in childhood cancer survivors compared to siblings [22,25]. Their diverse, and often age-specific needs are often unmet. [26] but include distress related to body image, fear, social isolation, and concerns about fertility [27]. Due to a host of psychosocial stresses following cancer, psychosocial guidelines recommend providing psychosocial interventions to survivors [28]. A wide variety of interventions exist to address long term physical and emotional sequelae of cancer including educational [29] cognitive behavioral [30], physical exercise [31–33], health behavior change [34] and social support programs. A number of nature-based programs exist, such as cancer camps and adventure therapy (usually for young adult cancer survivors) and are reviewed here [35]. Few programs explicitly evaluate the role of nature in healing.

org/research/funding-opportunities/ " (EAL, Principal Investigator and MJ and MCJ as co-investigators.) There was no additional external funding received for this study. The funders played no role in study design, data collection and analysis, decision on where and what to publish, or preparation of the manuscript.

**Competing interests:** The authors (EAL, CLO, MCJ and MJ) declare no competing interest. HRO is the founder and director of See You at the Summit, As such, she was involved in the process of the program evaluation, but the full group of authors has together made sure that reporting of results has been done thoroughly, openly, and without bias. This approach does not alter our adherence to all PLOS ONE policies on sharing data and materials.

**Abbreviations:** 3-Mo FU, 3-month follow-up after the program (referring to surveys); ATES, Adventure Therapy Experience Scale; CTCAE, Common Criteria for Adverse Events; day 9 FU, Final day of the program-Day 9 Follow-up (referring to surveys); HIPPA, Health Insurance Portability and Accountability Act of 1996; HRQoL, Health related quality of life; ICU, intensive care unit; MD, Medical doctor; NCI, National Cancer Institute; RN, registered nurse; SES, socioeconomic status; SYATS, See you at the Summit.

## Wilderness therapy has potential for adolescents impacted by cancer

Over the past decade there has been growing interest in the use of nature and wilderness therapy as a method of restoring wellness and promoting quality of life [36]. Nature therapies and "forest bathing" for adults have been described as reducing physiological and psychological symptoms of stress, decreasing anxiety, reducing depression and chronic pain symptoms and improving sleep and well-being [37]. A nature intervention increased well-being and reduced fatigue in adult cancer patients [38], and was considered the "most important" coping method in 2,355 adult cancer patients [39], A meta-analysis of 197 adventure therapy programs including 2,908 adults or children, showed the strongest benefits for clinical and self-concept outcomes [40].

Wilderness therapy has been used to address adolescent mental health distress with success. [41,42] with significant improvements in behavioral and emotional functioning [42]. However, previous outdoor and wilderness adventures for cancer survivors have mainly focused on young adult populations. Little is known about the feasibility, acceptability, and safety of wilderness therapy programs for adolescents with cancer. A recent scoping review of existing wilderness programs for childhood cancer survivors described increased social involvement, self-esteem, self-confidence, self-efficacy, social support, and physical activity [35]. The majority of these programs focused on adventure therapy, often in camps, and few described the impact of role of nature.

The primary aim of the study is to describe the feasibility, acceptability, and safety of See You at the Summit (SYATS) wilderness program. The secondary aim describes the impact on participants including a quantitative assessment of global and mental health as well as qualitative findings and observations describing the program participation experience of SYATS participants.

## Materials and methods

### Description of study design

This mixed-methods study had a pragmatic approach and collected data to assess the primary aim of feasibility, acceptability and safety of SYATS using criteria outlined by the National Center for Complementary and Integrative Medicine and Health [43]. The study used quantitative measures to document: success of recruitment, enrollment, and retention of the target population; adherence to the program; adverse events; program credibility, and data collection completeness. While program fidelity was examined, it was done not to ensure that the protocol was strictly followed but to allow for reflection on and improvement of the protocol. Participant observation and open-ended interviews were carried out to assess participation in and acceptability of SYATS component activities.

The secondary aim was examined using both quantitative and qualitative data methods. Standardized surveys were used to assess global mental and physical health outcomes in an exploratory way. Survey data was collected at baseline, at day nine follow-up (day 9 FU) and at three-months follow-up (3-mo FU). Pre- and post-data were compared and change scores are described from baseline to day 9 FU and baseline to 3-mo FU. This data should be used for future sample size calculations and for descriptive purposes only given the small sample size.

Participant observation and open-ended interviews were used to learn how program participation impacted participants. Interviews and field notes were transcribed, coded and analyzed in line with the methodology for qualitative content analysis as described by Graneheim and Lundman [44]. Thematic coding was used to identify words and phrases that are linked by a common theme [45].

## Wilderness program inclusion criteria

The study population (Table 1) included adolescents 13–17 years of age who were diagnosed with a cancer and were either in active treatment or finished treatment within the past two years and consented or assented (depending on age) to the nine-day wilderness experience in the summer of 2019. Participants were from Oregon and Washington, USA. They were a mean of 15.5 years old, 50% were female and all participants lived with both their mother and their father at the time of the program.

Staff were all volunteer and highly qualified, including experienced wilderness adventure leaders, health care providers, search and rescue teams, therapy dog trainers and people skilled in psychosocial care among others (Table 2 in S1 Appendix).

## SYATS program

SYATS is a nine-day structured wilderness-based backpacking program taking place in Oregon Deschutes National Forest, USA. SYATS was developed and lead by one of the authors

**Table 1. Description of study participants.**

| *Characteristics* | | |
|---|---|---|
| **Age (years)** | **Range 13–17** | **Mean = 15.5** |
| | **N** | **(%)** |
| Gender | | |
| Male | 4 | (50) |
| Female | 4 | (50) |
| Race | | |
| White | 4 | (50) |
| Asian | 1 | (12.5) |
| Multiracial (White + Black, Asian or Native American) | 3 | (37.5) |
| Living Setting | | |
| Rural | 1 | (12.5) |
| Suburban | 4 | (50) |
| Urban | 1 | (12.5) |
| Small town | 2 | (25) |
| Caregiver work | | |
| Full-time | 6 | (75) |
| Unemployed | 1 | (12.5) |
| Unable to work-caring for child | 1 | (12.5) |
| Caregiver-highest grade of schooling | | |
| Grade school | 3 | (37.5) |
| High School/GED | 5 | (62.5) |
| Post HS | 0 | |
| College | 0 | |
| Child's Insurance | | |
| Private | 7 | (87.5) |
| Medicaid | 1 | (12.5) |
| Which of the following describes the money situation in your household right now? | | |
| Comfortable | 4 | (50) |
| Enough but no extras | 3 | (37.5) |
| Have to cut back | 1 | (12.5) |
| Cannot make ends meet | 0 | |

(HRO). Program founder HRO has 18 years of experience leading wilderness trips. SYATS aims to restore a sense of independence, self-efficacy, and social support, improve global health and positive affect, and promote resiliency among the participants. Mechanisms for these improvements include structured peer interactions, promotion of independence and autonomy, a reduction in self-limiting beliefs, increased skills to bolster positive emotion, providing service to others, and participation in a wilderness experience. The program was structured with an introduction to wilderness skills, equipment, and exercises to promote team building, self-awareness, and independence. Teambuilding is used to support group of individuals to see themselves as part of a cohesive team, sharing tasks, goals and having healthy relationships. This is accomplished through team building exercises followed by reflection [46]. SYATS utilized teambuilding activities as part of the overall wilderness therapy program in order to enhance group cohesion.

SYATS includes hiking, camping, reflection (journaling, meditation, yoga), appreciation for nature, and social activities (campfire and group processes), leading up to a day summiting a peak. All participants medical records were reviewed by the SYATS medical team prior to acceptance into the program and SYATS followed a safety protocol. Participation in SYATS was free of charge and SYATS provided all equipment and clothing for teens to use. An overview of the program is given in Table 1 in S1 Appendix and further program description can be found here: https://seeyouatthesummit.org/.

## Human subjects ethics and consent process

Program recruitment took place through engaging key persons at hospitals in Oregon and Washington state who provided initial information about the program. After a teenager signed up for the wilderness program, the SYATS staff notified the teen and their guardian about the study and they were given the option to participate. Written informed consent for participation in the evaluation was obtained from guardians of those younger than 18, and for those age 18 and older. In addition, the study obtained written assent from minors for participation in the study. Each teen and their guardian were told that participation in the evaluation portion of SYATS was optional and would in no way impact their ability to be included in all aspects of the SYATS program. The consent and/or assent form included the following information: the goal of the study; what the participant will be asked to do; risks and benefits; compensation; confidentiality; and the voluntary nature of participation. The teenager and their guardian were given the opportunity to ask questions. A $100 educational stipend was provided to compensate participants for time they spent filling out questionnaires and completing interviews. Internal Review Board Ethics Committee For Legacy Health (Portland OR) approved the program and evaluation (Version 2.0_7_17_19, on 7/23/2019). No changes we made in the protocol following the IRB approval.

All records and information complied with the Health Insurance Portability and Accountability Act of 1996 (HIPPA) guidelines [47]. Participation and evaluation data are confidential and no identifying information was used in study documents or for publication. This study is reported in accordance with the Consolidated criteria for reporting qualitative studies (S2 Appendix).

## Measures and outcomes

**Quantitative data.** To assess feasibility, we evaluated recruitment efforts tracking individuals from first contact with SYATS to enrollment, and retention. Survey data on demographics, self-reported mental health and well-being, injuries or symptoms, and adventure experience were collected. While the staff did not a priori define expected adverse events, the

staff did keep detailed records of each event according to the National Cancer Institute (NCI) Common Terminology Criteria for Adverse Events (CTCAE) and included both serious and non-serious adverse events along with attribution. Each event was described including date, time, duration, differential diagnosis, any treatment provided, and resolution. Each event was assigned a severity score according to the CTCAE severity scale. Grade 1 Mild, Grade 2 Moderate, Grade 3 Severe, Grade 4 Life threatening or disabling, Grade 5 Death [11,12,48], by the medical staff. To assess acceptability, we asked participants whether they would recommend the program to friends (scale of 1–10) and tracked participation in the component activities using qualitative data.

Secondary outcomes related to program impact were evaluated using standardized surveys assessing mental and physical health. The following outcomes were assessed and PROMIS instruments are referenced. All used a 5-point Likert scale for response except pain which was ranked 0–10.

PROMIS Health related Quality of life (HRQoL) reflect symptoms over the past week and included these domains: physical functioning, anxiety, depression, fatigue, peer relationships, and pain interference. All scales were ranked on a 5-point scale ranging from "never" to "almost always" [49]. Physical functioning was measured using a 5-point Likert scale from "with no trouble" to "not able to do" [49–52]. PROMIS Positive affect was ranked on 5-point scale from "never" to "almost always" [53]. Mean scores were compared between baseline and day 9 FU or baseline and 3-mo FU.

Paper surveys were administered at the start of the program, on the final day (day 9 FU) of the program, and the survey was emailed or postal mailed to participants for the 3-mo FU. Patients were allowed to skip any questions.

**Qualitative data.** During the program, qualitative data was recorded in field diaries by a participant observer (MJ/*Male*) including unstructured observations on events, conversation and interactions between participants, and staff and volunteers. Participants in the program were aware that MJ was a part of the team as a participant observer. Participant observation is viewed as a key method in studying how people move, interact, and use space in social settings [54].

Interviews were performed with participants during the program, both in the form of daily unstructured ad hoc qualitative interviews corresponding to key events so as to capture the mood and thoughts of the participants in real time (Duration 1–20 minutes). Semi-structured exit interviews were performed at Day 9 FU. All interviews were recorded digitally (Duration 20–47 minutes, mean: 33 minutes). The interview guide covered areas such as; overall impression of program, experience of the activities, perception of staff and other participants, the nature experience, health aspects, safety, and motivation for participation (S3 Appendix). The exit interviews were performed by an external interviewer (*female*) who was an experienced wilderness therapist with support from MJ (*male*). The use of interviews as a data collection method allowed the interviewer and others to understand the inner experiences as expressed by the participant [55].

Staff held a debriefing meeting following the program. They described a number of experiences and their recommendations for future trips and the information was recorded in field notes by EAL.

## Data analysis

Feasibility was assessed using numbers and percentages for recruitment and retention. Quantitative outcomes related to mood and quality of life are described using percentages, means, and pre-post differences. We did not anticipate statistical differences between timepoints, but

data analysis was mainly explorative and used to demonstrate the ability to collect survey outcome data. In survey data, mean scores were evaluated between the pre and post phases. An overall program satisfaction question was delivered at day 9 FU and 3-Mo FU.

The qualitative analysis was performed in-line with the methodology as described by Graneheim and Lundman [44]. The analysis included all transcribed text data including ad hoc interviews, exit interviews and field diaries. Analysis of qualitative data involved both deductive and inductive approaches. The process moved back and forward but included the following steps: 1) Initially all text data was read several times for an overview on the content. 2) This was followed by a deductive approach where three of the authors (*EAL*, *MJ*, *and CLN)* coded data from a manifest standpoint (what the text says), where relevant meaning units (words, sentences, paragraphs) relating to aspects of feasibility and safety were extracted and sorted into subcategories, categories of feasibility, acceptability, and safety.

3) All data was read again and coded using an inductive approach in order to identify the manifest and latent content relating to program experience. 4) all content relating to a deeper latent meaning of the participants' program experience was identified, coded, and sorted into categories and interpretive cross-sectional themes (These are visualized in Table 5).

## Results

### Feasibility data, Aim 1, quantitative and qualitative findings

The in advance identified domains of feasibility as used in the deductive analysis referred to: ability to **recruit** the target population, **retention**, **adherence** to program, **safety assessments,** and **ability to collect both qualitative and quantitative data** on key outcomes. Finally, the study assessed whether the program was both **acceptable** and **credibl**e to participants.

SYATS program **recruitment** occurred primarily through three children's hospitals in Portland Oregon and Seattle Washington using word of mouth, postcards (one site) and a website that is publicly accessible (https://seeyouatthesummit.org/). The SYATS recruitment goals was eight participants (the Forest Service permit maximum group limit is 12, which includes 8 participants and 4 staff members. Hospital staff (nurses, social workers, and ambassadors) approached teens/families and introduced the wilderness program. A total of 21 teenagers, aged 12–17 were referred to, or expressed interest in participating in SYATS. Among this group, the cancer diagnoses included: brain tumor, leukemia, germ cell tumor, and lymphoma among others. Among the 21 with interest in SYATS, 4 (19%) were ultimately not eligible, 6 (35%) did not contact SYATS again, 3 (18%)were not interested or had a scheduling conflict and 8 (47%) signed up (See Table 2). Retention was 100% with all eight teens participating from day 1–9.

**Table 2. Tracking of teenagers who contacted SYATS as part of recruitment effort.**

| N = 21 (%) | Contacted or referred to SYATS |
|---|---|
| 4 (19%) | Not eligible: Significant illness or physical disability preventing participation (n = 3) or did not meet age criteria (< age 13) (n = 1) |
| | Outcomes among 17 (81%) eligible participants |
| 6 (35%) | No further contact, (% of eligible) |
| 3 (18%) | Not interested, said no, scheduling conflict, (% of eligible) |
| 8 (47%) | Signed up and went on trip, (% of eligible) |

Based on participant observation data, **adherence to the program** was high with 100% of the teens participating in the planned activities including hikes, lake swimming, camp chores, group teambuilding exercises, campfire discussions, mindfulness exercises, and journal writing. Staff **adherence to the program protocol** was also high, again following the outlined protocol, with a variety of skills and roles among staff members and an experienced leadership team (Table 2 in S1 Appendix) so that the protocol could be implemented.

Program activities appeared to be **acceptable to participants.** Three general quotes reflect that.

*This program meant the world to me. While I do hope that cancer ends, I hope this program continues until it does.*

(*participant 7*)

*This was the most amazing experience for me. I hope that many others get to go through the same journey I did and get the same positive results.*

(*participant 5*)

*I can climb a mountain. It's impossible not to build amazing friendships on the trail. I miss the trail snacks, river baths, campfires, tents immensely.*

(*participant 2*)

*I really enjoyed being in nature, the scenery really helped me connect spiritually and physically with myself. I hope that others get to enjoy an experience as amazing as mine. It was one of my favorite summer memories.*

(*participant 6*)

Engaging in new activities could be challenging.

*"I get anxious in a new experience. This [backpacking] is definitely different. Something I'm not used to, so I just wrote in my journal a lot. . .*

(*participant 6*)

Camp chores received the lowest enthusiasm initially, but as time passed, the participants grew to appreciate the need.

*"You run out of water, and that first drink of water is just awesome. You're rejuvenated because you've been out for how long. One day we were doing teambuilding and we were in the middle of the sun and I was out of water and I was stressed about not having water because it was hot and I was really thirsty. We finally took a break from that. . .. and got some water, and that was really nice."*

(*participant 4.*)

Program conditions including the wilderness setting, a mix of participant characteristics, and the camping arrangements were **assessed as being credible** by participants. The wide age range however lead to a reflection.

*"I think that the age differences are still playing a fairly large factor just because when you have a 13-year-old and an 18-year-old in the same group, there's going to be a bit of a*

*disconnect. They can still be cooperative and work together, but there's just going to be a difference in maturity. . .".*

(*participant 3*)

**Safety protocols** were carried out and **adverse events** were recorded. The safety protocol included an evacuation plan, the presence of medical experts and equipment, and reporting of adverse events. A helicopter was located at the nearest official Search and Rescue location and trained Search and Rescue experts were alert to the trip and participant needs. There was extensive medical equipment at base camp and in the field with participants. There were medivac trained staff (with wilderness-oriented medical rescue expertise and equipment) who hiked with the teens. Additionally, at base camp, all of the military crew had medivac experience and equipment (Table 2 in S1 Appendix) Participants were given all prescribed medications by the staff medical doctor (MD) and a second nurse signed off on each administration.

The safety protocol dictated that an Adverse Event form be used if an event occurred. The protocol dictated that serious adverse events would be reported to the Legacy Institutional Review Board in accordance with their reporting guidelines as well as to the participant's primary healthcare provider of record for the trip. Provisions (ground and air transport) were made in the case of a needed evacuation.

Ultimately, no serious adverse events occurred. A number of mild events were reported typical of the type of events that can occur on a wilderness trip (See Table 3). Events required little to no intervention to resolve.

A teenager reflects on his physical capacity

*"I feel like I've done a lot better than I expected I would. I feel definitely it's not easy, but I've been able to do it without too much struggle. . ..I had a headache two days ago, so that's pretty much as far as the pain for what I've done goes. I mean I've got a little feet pain, back pain, but I'm sure everybody does."*

(*participant 4*)

**Acceptability and completion of study assessments** was high. One hundred percent of the survey data was collected at baseline and at the 9 Day FU. At the 3-mo FU survey data was missing from 12.5%, 1 participant missed the whole survey and another missed the final two scales (positive affect and the Adventure Therapy Scale). Outcome survey assessment tools, qualitative interviewing, and participant observation used for evaluating outcomes were perceived (by staff) as easy to deliver and easy to engage in by the participants.

As the first SYATS trip, one goal was to evaluate the most effective way to implement the protocol, as outlined, and to assess the strengths and weaknesses of different approaches, schedules, education timing and content, and group and training exercises outlined in the protocol (see Appendix A). Based on participant observation, on day 1, additional support for logistics would have been helpful as this was a busy time when participants received gear, learned basic backpacking skills, and were getting to know each other. Dedicated logistic staff might have allowed for additional group exercises to be carried out. Day 2, 4–7 were carried out as planned. On Day 3, additional rest time occurred in response to participant needs. On Day 8, rain delayed some activities and on Day 9, exit interviews took more time and the teens were not able to cook breakfast.

**Table 3. CTCAE adverse health events.**

| Adverse health events | Grade* 1–5 | Action taken (by whom) | Duration | Attribution** | Cause |
|---|---|---|---|---|---|
| *Verifiable complaints* | | | | | |
| Dermatology: Skin breakdown/ Small cut/abrasion on finger (n-2) | 1 | RN, cleaned, band aid applied | | Definite | Small cut/abrasion on finger (n = 1) Small cut while shaving (n = 1) |
| Dermatology: Skin breakdown (n-2) | 1 | Monitor feet more often, tape, | 2 days | Definite | Blisters on feet, minor, due to hiking |
| Pruritus (n-1) | 1 | RN Monitoring | | Definite | Mosquito bite with minor swelling |
| Epistaxis (n-1) | 1 | MD performed temporary tamponade | 2 hours | Unlikely | Participant stated, 'she gets nose bleeds at home' |
| *Subjective complaints* | | | | | |
| Gastrointestinal discomfort (n = 1) | 1 | MD aware: Monitoring | <12 hours | Possible | |
| Constipation (n-1) | 1 | MD aware | 48 hours | Possible | Suspected anxiety over going to the bathroom in the woods. |
| Headache (n = 2) | 1 | MD aware. administered meds as needed. | Intermit-tent responded to treatment | Unlikely | (n-1) history of headaches due to cancer diagnosis. (n-1) unclear, participant stated "I get frequent headaches" |
| Rhinitis (n-2) | 1 | MD Monitoring, administering meds as needed | Intermit-tent | Likely | (n-1) Participant had allergies at baseline. |
| Gastrointestinal discomfort (n = 1) | 1 | MD administered prescribed Ondansetron preventatively | | Likely | Participant reported to MD too much sugar intake from her birthday cake may make her stomach upset. |
| Knee pain (n-1) | 1 | MD assessed, Rest | < 2 hours, participant resumed hiking | Definite | Participant noted knee tenderness while hiking. |
| Ankle pain (n-1) | 1 | RN assessed, removed participant's backpack. | discomfort went away within 1 hr. | Definite | Participant hiking downhill and mis-stepped, noting sudden discomfort. |

*Grade: 1 Mild AE, 2 Moderate AE, 3 Severe AE, 4 Life threatening or disabling, 5 Death related to AE [56].

**Mild**—Events required minimal or no treatment and did not interfere with participants' daily activities.

**Moderate**—Events result in a low level of inconvenience or concern with the therapeutic measures. Moderate events may cause some interference with functioning.

**Severe**—Events interrupt a participant's usual daily activity and may require systemic drug therapy or other treatment. Severe events are usually potentially life-threatening or incapacitating.

** AE Attribution: Unrelated, unlikely, possible, probably, and definite. AE can be related to underlying disease, medications, or the program.

**Program impact, aim 2, quantitative findings.** **Program impact** was evaluated in an exploratory way assessing Global Health, including physical health, anxiety, depression, ability to participate in social roles and activities, peer relationships, fatigue, and pain interference and intensity as well as assessing positive feelings, and satisfaction with the program and its execution (Table 4).

While the small sample size precluded meaningful significance testing, PROMIS baseline data showed small improvements in the majority of outcomes at day 9 FU and 3-mo FU. As can be the case with small numbers, mean changes can be strongly influenced by a few persons. At day 9 FU, there were improvements in physical functioning, anxiety, depression, fatigue, and peer relationships. There was no change in positive affect and pain was increased.

At 3-mo FU, physical functioning again decreased as did anxiety, fatigue, and pain interference compared to baseline. Depression was increased reversing improvements seen at day 9 FU. Positive affect improved and overall pain was decreased compared to baseline.

Overall satisfaction with the trip was high (9.25) at day 9 FU and slightly higher at 3-mo FU (9.67).

**Table 4. Baseline, Day 9 FU and 3-Mo FU, mean scores for each subscale, difference in mean scores between baseline and day 9 FU, and between baseline and 3-mo FU.** Bolded numbers indicate change in desired direction.

| PROMIS Measure (response range) | Baseline mean scores (range of means) | Day 9 FU mean scores (range); | Difference (and range) of Mean score between baseline and day 9 FU | 3-mo FU mean scores (range) | Difference (and range) in mean score between baseline and 3-mo FU (range of diff) |
|---|---|---|---|---|---|
| Physical functioning (1–5) | 4.56 (3.5–5) | 4.66 (3.5–5) | **0.09** (0.0–0.50) | 4.61 (3.75–5.0) | **0.05** (-1.75–3.5) |
| Anxiety (1–5) | 2.22 (1–3.5) | 1.91 (1.25–5.0) | **-0.31** (-1.75–1.0) | 2.08 (1.25–5.00) | **-0.14** (-1.75–3.50) |
| Depression (1–5) | 1.91 (1–4) | 1.56 (1.0–3.25) | **-0.35** (-1.25–0.0) | 2.46 (1.0–5.0) | 0.56 (0–3.25) |
| Fatigue (1–5) | 2.09 (1.0–3.25) | 1.63 (1.0–3.0) | **-0.46** (-1.5–0.25) | 1.71 (1.0–3.0) | **-0.38** (-1.25–0.25) |
| Peer relations (1–5) | 4.16 (3.25–5) | 4.38 (3.25–5.0) | **0.22** (-0.25–1.25) | 4.14 (2.75–5.0) | -0.01 (-1.25–0.50) |
| Pain interference (1–5) | 2.0 (2.75–5.0) | 2.06 (1.0–3.25) | **0.06** (-1.25–0.75) | 1.61 (1.0–3.25) | **-0.39** (-1.0–0) |
| Positive affect (1–5) | 3.91 (3.0–5.0) | 3.91 (2.75–5.0) | 0.00 (-0.5–0.75) | 4.29 (3.5–5.0) | **0.39** (-0.5–1.00) |
| Overall pain (0–10) | 2.5 (1.0–5.0) | 3.13 (1–6) | 0.63 (range -2–4) | 1.86 (0–4.0) | **-0.64** (range -2–0) |
| | | Day 9 FU | | 3-mo FU | |
| Considering your complete experience with See You at the Summit, how likely would you be to recommend it to a friend? (0–10) | *Not asked at baseline by design* | 9.25 (range = 8–10) | | 9.67 (range = 8–10) | |

Each PROMIS scale has 4 questions with a scale of 1–5. Higher scores indicate more problems except for physical functioning, peer relations and positive affect where high scores are better. Positive differences between baseline and follow-up surveys indicate improvement. For all other outcomes low scores are most desirable and negative differences between baseline and follow-up surveys show improvement. Some differences show rounding errors.

## Program impact-aim 2, qualitative findings

Overarching, central findings from the inductive qualitative analysis are based on the participants' experiences of self-confidence and peer connection which developed as participants experienced physical competency, group leadership, and personal strength. This is reflected here.

*"I hope to take back that I accomplished this because I usually have a 'can't do attitude,' and I just climbed a mountain!"*

(*participant 7*)

*"This has definitely built my confidence quite a bit, so that was very nice. It makes me feel happy. It just feels good that I don't doubt myself so much because I do that quite often after treatment."*

(*participant 1*)

*"I mean, I was worried about not being capable to be in the front or make sure everyone else was okay. My partner leader got sick halfway, and I had to lead without him and that was even more stressful which was kind of crazy. In the end, it worked out, and you were like, wow, I just led people up a mountain, so that was pretty cool. It's really rewarding and it's satisfying to see. You know that you did that."*

(*participant 2*)

The development of peer connection is expressed by a participant:

*". . .knowing each other and knowing what we've gone through because it's just nice to know we're not the only ones."*

(*participant 1*)

In the inductive latent qualitative analysis, four core themes were identified; a) inner growth and experiencing a sense of accomplishment; b) teambuilding as a means of bonding and individual and group development; c) staff support—their constant presence and balanced interference and d) being in and connected to nature as a supportive environment. Below, illustrative quotes are shown that also represent the categories (*group processes*, *personal development*, and *backpacking and camping skills)* that exists within the themes, (Table 5).

**a) Inner growth and experiencing a sense of accomplishment.**   A teenager describes how the trip led to an emotional turning point. She said:

*". . .don't remember the last time I had a breakdown like that. After cancer, I was, like, I don't have anything to cry about. I don't want to think about those sad feelings. So for months now, I've probably just been bottling it up and I didn't even realize it. When we were just hiking and people started getting on your nerves, and you're tired and hungry, and my shoulders are just killing me, and we weren't really that close. . .and yeah that physical deterioration just led into that emotional pop."*

(*participant 2*)

Following the cathartic moment, it was easier to connect with the others in the group. She continued:

*"After I had all that emotion off, it was like why would I be grumpy? So I just shrugged it off. When people came by, I would sit up, I would talk to them more, I would joke around, sing songs, yeah. It was just much easier for me to be around everyone."*

(*participant 2*)

One participant had an insight on her own social behaviors in the context of a group. She described herself as talking a lot especially when nervous.

**Table 5. Cross-sectional themes and categories with examples of coding.**

| Cross-sectional themes | Categories | | |
|---|---|---|---|
| | *Group processes* | *Personal development* | *Backpacking and camping skills* |
| **a) Inner growth and experiencing a sense of accomplishment** | Sharing and opening to one's vulnerability and sense of weakness—stimulate group cohesion and bonding with peers<br>"Breakthroughs" as a basis for reflection and connection | Pushing limits realizing one's own capacity<br>Developing emotional strength<br>Realization of being normal and like anyone else<br>Being able to accept and have fun with others, although they might not be friends in another setting | Developing skills—Learning and finding joy, accomplishment and acceptance in the camping chores needed |
| **b) Team building—as a means of bonding and developing as individual and as a group** | Learning to attend to needs of others | Being"leader of the day" promotes individual insight and learning | Doing trail and camping activities together provides a space for bonding |
| **c) Facilitator support—a constant presence and balanced interference** | Facilitating and providing support in group and individual"breakthroughs" | Facilitators—being fully present and seeing the need for emotional support—promoting reflection | Facilitators—providing knowledge, means and balanced guidance for supporting learning |
| **d) Being in and connected to Nature as supportive environment** | Togetherness—reaching the summit as a team<br>"Free leisure time" stimulating group cohesion | Finding peace and calmness that promotes reflection<br>Learning how our actions affects impact on nature<br>Seeing and appreciating the beauty of small and big things in nature | Learning camping skills and leave no trace principles<br>Learning to meet challenges and exceed one's perceived physical and mental limitations<br>Feeling physically stronger |

*"I feel more acceptive of people, what they have to offer. I'm working on not talking a whole lot, and this really helped. . ..I learned that people have things to say too."*

(*participant 6*)

Cancer can result in social isolation and social challenges for teens at home or with peers [57]. At SYATS, the shared experience of cancer provided a common ground.

*"We've all been through tough times. . .at school there's a lot of people and you're not sure if they care about you so you always have a wall with certain people. But with these [SYATS] people I felt I was able to let down my wall."*

(*participant 6*)

The feeling of developing emotional strength is exemplified by one of the teenagers reflected on a lesson she will bring home from the wilderness trip.

*"I think now I won't let certain things be a block or obstacle as much. Even if it is, I want to tackle it more with the strength I've found during this last week. Mental wise, I think that perspective will always follow you around. IF it's a positive mindset, it will affect you positively because that'll dictate how you act and think."*

(*participant 8*)

Another teenager describes how she had become happier during the trip.

*"I felt that I became happier because cancer kind of closed me off a little bit because I couldn't be out at school and everything because of my compromised immune system. It really got me to the point where I'm with other kids who know my issues and it was really nice. . ..just to get somewhat normal again with kids who are all in situations similar to mine."*

(*participant 7*)

## b) Teambuilding—As a means of bonding and developing as an individual and a group.

Teambuilding exercises were used throughout the program to enhance peer support, communication, and trust.

*"After we failed to complete the pole activity, we had a conversation. First, we went and journaled about how we felt about the group, how we felt the group was impacting you, how you were impacting the group. Then we shared some of those feelings with each other, and I think that really helped make the group more cohesive, being more honest with each other."*

(*participant3*)

On days of rest and recovery participants were observed taking time for camping chores, bathing, and washing their clothes. During this non-facilitated time, they were able to talk with each other on their own terms, exchanging moments of their cancer experience; losing hair, being tired and isolated during treatment, and their sense of being different from other teenagers. Being in nature, swimming in pristine lakes, laughing and talking, the setting gave the

opportunities for calmness and relaxation supporting them in opening up to each other. During the hikes as well as in the camps there was space for the teenagers to get to know each other better. While fetching clean water, washing, or during the process of digging latrines, conversations and stories about experiences in daily life, school, cancer treatment procedures, or hair loss happened naturally. One of the teenagers described her pride in cooking dinner with another participant:

*"One night we had pizzas and you have this tiny little fire to do it on, and me and my friend. . .did dinner that night. And that was so much work. . . but then everyone was sitting around the fire eating yummy pizzas. That was so fun because you're like 'wow we did all that work,' and this tastes really good."*

(*participant 2*)

**c) Staff support—A constant presence with balanced interference.**   The program director and staff were observed discussing where and when to step-in in order to provide participant support. They were aware that doing "just enough" can support and strengthen self-esteem but doing "too much" can be negative. In the beginning of the event participants were observed to ask staff directly for practical support (trekking and camping skills such as how to pack and adjust backpacks, pitch the tent, cook food etc.). Emotional support developed in a subtle way as bonds formed among the full group (staff and participants). A participant shared an experience with a staff member when she was feeling low:

*". . . I wasn't crying or doing anything. I was just sitting down. She asked me how I was doing, and I said I'm fine. Just like I say that every day. She was like you're not fine, and you know that first sign of recognition that they're not fine. She said, 'what's under that fine?' And I just broke down and was like oh no, she saw past me. Every day after that we became closer and would joke around, and she's a great mentor. . .That was pretty incredible to know she recognized that."*

(*participant 2*)

**d) Being in and connected to nature as supportive environment.**   The program was based in nature. Participants described both feelings of not being comfortable in this unfamiliar setting and how nature supported them to develop group cohesion, to grow on a personal level, developing personal insight and learning new forms of skills necessary for being on the trail. A participant reflected on this:

*"It just feels amazing, and that really sank in when we were at the summit looking at the Three Sisters, and Little Brother, and some other mountain I forgot the name of. Also the lake below was just gorgeous. Just all these mountains. . . that we could see. Just so beautiful. It was amazing to take it in. We all got there as a team."*

(*participant 1*)

The summiting experience included a sense of awe by another participant:

*"It was a little grueling going up Broken Top, but once we finally reached the saddle, the highest point that we could hike up, it was an incredible experience looking out across the view you*

*could see from the ridge and seeing No Name Lake in the caldera of the volcano. That's just such a pristine lake. So clear with just the wind creating little ripples and the waves."*

(*participant 3*)

The participants also relate to a feeling of being physically stronger as a result of being in nature:

*"Physical health-wise I feel a lot stronger. I feel a lot more capable of doing stuff. Since after or during treatment people think you're not capable of doing a certain thing, so they leave you out, and I can say I've hiked this much."*

(*participant 4*)

Another participant elaborated on how hiking affects the feeling and perception of physical strength:

*". . .coming out here and really seeing how much better I am than how I was in treatment, and walking around the hospital got me out of breath, coming out here and being able to carry 30+ pound backpack it's amazing to feel my body have that much strength again."*

(*participant 3*)

## Discussion

Through a medically supported nine-day backpacking trip highlighting the summiting of one of Oregon's peaks, teenagers connected to peers living the same challenges and cultivated self-confidence and emotional resiliency. The combination of immersion in nature, connection with other teenagers, and the experience of achieving wilderness challenges encouraged a change in how adolescents viewed themselves, their illness, and their future. This evaluation adds to evidence that a wilderness program for teens impacted by cancer is feasible and acceptable and that it can be implemented in a safe way. A larger randomized controlled trial is needed to understand more about effectiveness of nature-based wilderness trips on mood, fostering resiliency and autonomy, and preventing and reducing depression and anxiety that can follow a cancer diagnosis and treatment.

The SYATS program provides preliminary data that is in accordance with data from other outdoor or adventure therapy programs including increased social involvement, increased self-esteem, self-confidence, self-efficacy, social support, and physical activity [35]. This program evaluation fills a gap where few services or research projects have examined the use of nature and wilderness experiences among adolescents facing cancer. In a scoping review (by current authors, EAL, MJ, HRO and MCJ) [35] most nature-based programs focus on young adults or children's camps, but not on teens. Previous programs were most likely to be an adventure therapy program (73%), did not specifically evaluate the role of nature (100%), did not report on the training and qualifications of staff (33%), did not provide information on safety issues in wilderness programs (87%), and where information on race/ethnicity was reported 78% of enrollees were white. Adolescents are a particularly vulnerable population at risk for the psychosocial sequelae following treatment for cancer and are an important group for the development of future programs, yet they are rarely the participant population in these programs.

Survey, interview, and observational data showed promising trends in the direction of healing and resiliency, reduced isolation, independence and autonomy, the development of

outdoors skills, and the development of skills to bolster awareness. The use of gratitude processes, mediation, yoga, and journaling can foster a sense of inner quiet and reflection and were enjoyed by the participants. Morning check-ins helped participants to set goals and make conscious choices. Hikes, and summiting a peak fostered a sense of accomplishment (overcoming a physical challenge) and built a sense of personal meaning. Including a daily hike leader built a sense of leadership, confidence, and competence. The protocol dictated that the most challenging day—summiting a peak—would be achieved by the entire group together (or not). This philosophy helped foster inclusion, taught group dynamics, and taught the "leader of the day" to have patience with, and sensitivity to, different abilities.

As expected, small numbers limit conclusions about the program impact [58], but the general trend in improvements is promising. Mean score differences over time are best used for power analysis calculations for a future controlled trial. Typical of a small study, in the case of anxiety, four participants showed a decrease and two reported a small increase making it appear that there were few differences over time. The two who reported a small increase had reported almost no anxiety at time one and it may be that after the program, they were better able to identify moments when they felt worried. A larger study with a more rigorous design is needed to provide conclusive findings on possible effectiveness of a program such as SYATS. Future wilderness programs for teen cancer survivors and the introduction of a control group could provide more definitive data to answer the key program questions. Furthermore, survey data collection at later time points (up to one year) is strongly recommended to establish the lasting impact of a program. The mixed methods design that also provided data through participant observation, ad hoc interviews, and exit interviews provided rich data in a way that survey data cannot. Data was rich and explanatory, and the author team perceived that the "information power" of qualitative data as shown in the results section is sufficient to meet the objective of the study, as recommended by Malterud et al [59].

## Lessons learned

There are many lessons to be learned from the STATS program. A wilderness program protocol necessitates adaptation. For instance, some planned activities were eliminated or reduced when participants felt tired after strenuous hikes or when staff needed to prioritize meeting participants' basic needs. In other cases, a given hike took longer than expected or there were weather events and subsequent activities were shortened, delayed, or skipped. The original protocol included more activities than could be carried out on some days. Camp chores (setting up and breaking down camp, making food, pumping water) needed to be prioritized over other proposed elements at times. However, every effort was made to use the protocol as the outlined structure of activities. Staff and participants were able to carry out most key activities as planned each day with some adaptations.

Given the vulnerability of participants, thorough descriptions of safety precautions and adverse events are critical. The data from the present evaluation shows that such a program can be carried out safely with only minor adverse events, most consistent with wilderness activities. Slightly higher pain scores following the program may be related to the long days of hiking and the participants' relative inexperience in the outdoors.

The study team concluded that recruitment was possible even in this young and medically vulnerable population. Personal relationships with staff at cancer treatment centers was key for building interest and trust in the SYATS program and thus, contact with the eligible population. It is hoped that documentation of the safety protocol and adverse events reporting will encourage confidence in referring medical staff and families for future trips.

Data collection was comprehensive and complete. Participant observation and open-ended interviews triangulated with survey data provided a well-rounded perspective on program fidelity and success. In the future, data collection should be carried out using a web-based survey or telephone interview in advance and after the program in order to 1) avoid possible effects of program-related expectation-anxiety and/or immediate post-trip enthusiasm, and 2) avoid taking up time that could be used for other program components.

Nature appreciation emerged in the moment and in the interactions, during quiet moments, and when contemplating views. Knowing that this element was strongly experienced, future trips might focus on promotion of quiet time in nature, time appreciating views of nature, and gratitude exercises related to nature. Interestingly, based on the previous scoping review [35] most previous pediatric cancer wilderness and camp programs did not specifically promote nature, but instead nature was taken for granted as a backdrop [35]. This evaluation is one of the first wilderness or cancer camp programs to collect specific data on the impact of nature in healing for teens with cancer.

The PROMIS global and mental health scales have strong validity and are easily implemented in this population. Their use is recommended for future wilderness programs given their ease of implementation and strong validity. However, in this small group, we caution against over interpretation of mean PROMIS sub-scale scores given the small sample. Use of multimethod assessments provided rich data to better understand where participants stand in relation to norms (PROMIS) and to understand the lived experience.

## Conclusions

Participation in the SYATS wilderness program for adolescents who were in or just finished with cancer treatment was feasible, acceptable, and safe among this medically vulnerable population. Qualitative findings suggest development of self-confidence and peer connections. The program was evaluated as increasing a sense of personal accomplishment, supporting social interaction, having strong staff support, and capitalizing on the natural surroundings. Larger randomized controlled studies would greatly advance the field and are needed to learn whether similar programs can improve longer-term psychosocial outcomes.

## Supporting information

**S1 Appendix. Tables 1–3 Provides more detailed information regarding program activities, staff qualifications and roles, and supplementary quotes.**
(DOCX)

**S2 Appendix. COREQ report.**
(PDF)

**S3 Appendix. Interview guide.**
(PDF)

## Acknowledgments

The authors want to acknowledge the incredible group of young people impacted by cancer who participated in the SYATS program and its evaluation. We are grateful for the considerable time and energy they invested in sharing their experience with us.

## Author Contributions

**Conceptualization:** E. Anne Lown, Heather Rose Otto, Miek C. Jong, Mats Jong.

**Data curation:** E. Anne Lown, Mats Jong.

**Formal analysis:** E. Anne Lown, Heather Rose Otto, Christine Lynn Norton, Miek C. Jong, Mats Jong.

**Funding acquisition:** E. Anne Lown, Heather Rose Otto.

**Investigation:** E. Anne Lown, Mats Jong.

**Methodology:** E. Anne Lown, Mats Jong.

**Project administration:** Heather Rose Otto.

**Resources:** E. Anne Lown, Heather Rose Otto.

**Validation:** E. Anne Lown, Heather Rose Otto, Christine Lynn Norton, Miek C. Jong, Mats Jong.

**Visualization:** E. Anne Lown, Miek C. Jong.

**Writing – original draft:** E. Anne Lown, Heather Rose Otto, Mats Jong.

**Writing – review & editing:** E. Anne Lown, Heather Rose Otto, Christine Lynn Norton, Miek C. Jong, Mats Jong.

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
