## [Decision Letter · Decision Letter 0]

10 Apr 2023

PONE-D-22-28734Program evaluation of a wilderness experience for adolescents facing cancer: A time in nature to heal, connect and find strengthPLOS ONE

Dear Dr. Jong,

Thank you for submitting your manuscript to PLOS ONE. After careful consideration, we feel that it has merit but does not fully meet PLOS ONE’s publication criteria as it currently stands. Therefore, we invite you to submit a revised version of the manuscript that addresses the points raised during the review process.

We look forward to receiving your revised manuscript.

Kind regards,

Cho Lee Wong, PhD

Academic Editor

PLOS ONE

Journal Requirements:

"HRO – Funding for the See You at the Summit program was provided in part by the OHSU Knight Cancer Institute Community Partnership Program. URL: https://www.ohsu.edu/knight-cancer-institute/community-partnership-program-grants

EAL - Funding for the Evaluation was provided by the University of California San Francisco, Osher Center for Integrative Medicine, Integrative Oncology Research Pilot Award. URL: " ext-link-type="uri" xlink:type="simple">https://oshercenter.org/research/funding-opportunities/"

"HRO – Funding for the See You at the Summit program was provided in part by the OHSU Knight Cancer Institute Community Partnership Program. URL: https://www.ohsu.edu/knight-cancer-institute/community-partnership-program-grants

EAL - Funding for the Evaluation was provided by the University of California San Francisco, Osher Center for Integrative Medicine, Integrative Oncology Research Pilot Award. URL: " ext-link-type="uri" xlink:type="simple">https://oshercenter.org/research/funding-opportunities/"

6. Thank you for stating the following in the Competing Interests section: 

"EAL, CLO, MCJ and MJ: The authors have declared that no competing interests exist.

HRO: I have read the journal's policy and the authors of this manuscript have the following competing interests: Founder and director of See You At The Summit, the program that has been evaluated. As co-author I have been involved in the full process of the program evaluation, but the full group of authors has together made sure that reporting of results has been done openly and unbiased."

7. In your Data Availability statement, you have not specified where the minimal data set underlying the results described in your manuscript can be found. PLOS defines a study's minimal data set as the underlying data used to reach the conclusions drawn in the manuscript and any additional data required to replicate the reported study findings in their entirety. All PLOS journals require that the minimal data set be made fully available. For more information about our data policy, please see http://journals.plos.org/plosone/s/data-availability.

Additional Editor Comments:

Dear Authors

Thanks for the opportunity to review this manuscript. The topic of this manuscript is interesting and fills a gap where few services or research projects have examined adolescents' use of nature and wilderness experiences. I just have a few small comments for your consideration:

1) Line 83: Please specify the type of chronic disease

2) Line 93: Please elaborate on previous programme that addressed psychosocial challenges and explain why wilderness therapy was chosen.

3) Line 147: Why money situation evaluated in the baseline?

4) Line 153: It is suggested that Table 2 be placed in the supplementary file rather than the main text.

5) Line 292: It is recommended to present the contents of Table 3 in text.

6) Line 384: Table 5, please add the actual mean score along with range

7) In the discussion, suggest to compare the content and results of the current wilderness programme with previous programme.

Reviewers' comments:

Reviewer's Responses to Questions

**Comments to the Author**

1. Is the manuscript technically sound, and do the data support the conclusions?

Reviewer #1: Yes

Reviewer #2: Yes

Reviewer #3: Yes

Reviewer #4: Yes

Reviewer #5: Yes

2. Has the statistical analysis been performed appropriately and rigorously? 

Reviewer #1: Yes

Reviewer #2: Yes

Reviewer #3: Yes

Reviewer #4: Yes

Reviewer #5: N/A

3. Have the authors made all data underlying the findings in their manuscript fully available?

Reviewer #1: Yes

Reviewer #2: No

Reviewer #3: Yes

Reviewer #4: Yes

Reviewer #5: No

4. Is the manuscript presented in an intelligible fashion and written in standard English?

Reviewer #1: Yes

Reviewer #2: Yes

Reviewer #3: Yes

Reviewer #4: Yes

Reviewer #5: Yes

5. Review Comments to the Author

Reviewer #1: 1-The authors should explain about how to screen the qualilative data and should show the data that related or not related between participants.

2-In discussion part the authors should clearly describe why SYATS program that effect in participant.

Reviewer #2: Overall, a strong piece of writing that might use some tweaking in terms of flow and presentation.

The authors may also provide research implications.

Overall, the writing is strong, but the flow and presentation could be improved.

The authors may also provide research implications.

Reviewer #3: Program evaluation of a wilderness experience for adolescents facing cancer: A time in nature to heal, connect and find strength. This manuscript good presentation, Overall, this paper has clear research issues, appropriate study methods, and presents interesting results.

Reviewer #4: Program evaluation of a wilderness experience for adolescents facing cancer: A time in nature to heal, connect and find strength. Overall clear, concise and interest to reader. Its interesting for publish jouraal. accept

Reviewer #5: The authors have provided a detailed description of a program of a wilderness experience for adolescents with cancer. the intent is noble and good- and the program is described discursively in a study of 8 participants, which really constitutes a preliminary/feasibility study. The small number of subjects is not really analyzed- nor evaluated in a systematic way beyond the success of participation.

the manuscript describes the success of the project and provides ample- perhaps too many detailed quotations. the manuscript could benefit greatly from substantial shortening- and moving many of the anecdotal quotes to a series of supplementals documents organized by the questions asked- and the responses. as presented, it is repetitive and works against its purpose- a demonstration of the feasibility and utility of a program- that could be extended in scope. the numbers are too small to establish any rigorous statistical analyses.

The final sentence of the abstract offers a vague statement that should be reworded. "controlled" studies is confusing and should not imply a randomization or standard clinical study for a handful of outcomes. Please be careful here and consider the next steps- is it continue the program or is it to develop clear and statistically compelling design- which would be very hard given the heterogeneity of patients, treatment schedules and timing of the structured experiences.

6. PLOS authors have the option to publish the peer review history of their article (what does this mean?). If published, this will include your full peer review and any attached files.

Reviewer #1: No

Reviewer #2: No

Reviewer #3: No

Reviewer #4: No

Reviewer #5: No

---

## [Author Response · Author response to Decision Letter 0]

30 May 2023

Information is already added in the file "Response to reviewers" but repeated here:

RESPONSE TO REVIEWERS

PONE-D-22-28734

Program evaluation of a wilderness experience for adolescents facing cancer: A time in nature to heal, connect and find strength

PLOS ONE

Journal Requirements:

- Corrected

- It is filled out and included as an attachment with the resubmission.

-In doing research with vulnerable participants, especially children and adolescents with cancer it is crucial that we adopt highly ethical methods, which we think we employed. In section 2.4, lines 182-184 of our submitted version we believe we addressed this issue, but we added one additional phrase for clarity (underlined): “Written informed consent for participation in the evaluation was obtained from guardians of those younger than 18, and for those age 18 and older. In addition, the study obtained written assent from minors for participation in the study (Supplement 2).”

"HRO – Funding for the See You at the Summit program was provided in part by the OHSU Knight Cancer Institute Community Partnership Program. URL: https://www.ohsu.edu/knight-cancer-institute/community-partnership-program-grants

EAL - Funding for the Evaluation was provided by the University of California San Francisco, Osher Center for Integrative Medicine, Integrative Oncology Research Pilot Award. URL: https://oshercenter.org/research/funding-opportunities/"

- See below under point #5.

"HRO – Funding for the See You at the Summit program was provided in part by the OHSU Knight Cancer Institute Community Partnership Program. URL: https://www.ohsu.edu/knight-cancer-institute/community-partnership-program-grants

EAL - Funding for the Evaluation was provided by the University of California San Francisco, Osher Center for Integrative Medicine, Integrative Oncology Research Pilot Award. URL: https://oshercenter.org/research/funding-opportunities/"

-Thank you for helping us to clarify the above. We have added relevant content related to funders in our cover letter. Thank you for making the relevant changes online.

6. Thank you for stating the following in the Competing Interests section: 

"EAL, CLO, MCJ and MJ: The authors have declared that no competing interests exist.

HRO: I have read the journal's policy and the authors of this manuscript have the following competing interests: Founder and director of See You At The Summit, the program that has been evaluated. As co-author I have been involved in the full process of the program evaluation, but the full group of authors has together made sure that reporting of results has been done openly and unbiased."

- This content has been added to the statement of competing interests in our cover letter. Thank you for making the change online.

7. In your Data Availability statement, you have not specified where the minimal data set underlying the results described in your manuscript can be found. PLOS defines a study's minimal data set as the underlying data used to reach the conclusions drawn in the manuscript and any additional data required to replicate the reported study findings in their entirety. All PLOS journals require that the minimal data set be made fully available. For more information about our data policy, please see http://journals.plos.org/plosone/s/data-availability.

- We have contacted PlosONE and received an exception on the submission of the dataset to a public repository due to the small number and the extensive photo and videographic documentation of the trip, making it easier to identify participants. We have however a clean data set that we will maintain and that can be used by qualified researchers. We have outlined the process for access to the data in the Cover letter to the editor. As well as in the appropriate section of the online submission system

Additionally, please revise text regarding qualitative data as follows: 

“Data from observation and qualitative interviews contains sensitive information and is available upon request from the medical director, Dr Joshua Hurwitz joshua.j.hurwitz@gmail.com, and after approval from the Internal Review Board Ethics Committee For Legacy Health (Portland OR, USA)”

- Supporting information files have been updated and in-text citations have been updated to match accordingly. 

- All our citations are valid and no changes has been made since previous version.

Additional Editor Comments:

Dear Authors

Thanks for the opportunity to review this manuscript. The topic of this manuscript is interesting and fills a gap where few services or research projects have examined adolescents' use of nature and wilderness experiences. I just have a few small comments for your consideration:

1) Line 83: Please specify the type of chronic disease. 

- We provide additional information for the reviewer below to this important and common issue of long-term late effects to the health of young cancer survivors. In the effort to contain the length we have added a single sentence to the manuscript to provide some more information, underlined below. 

Line 84: “At least 66% of child and adolescent cancer survivors have one or more chronic diseases as defined by the Common Terminology Criteria for Adverse Events (version 3) (39) including conditions such as congestive heart failure, major joint replacement, second malignant neoplasms, severe cognitive dysfunction, coronary artery disease, or renal failure.(11)” 

For the reviewers: Data was collected from 10,397 long-term cancer survivors and 3034 of their siblings on chronic health problems including congestive heart failure, major joint replacement, second malignant neoplasms, severe cognitive dysfunction, coronary artery disease, cerebrovascular accident, renal failure or dialysis, uncorrectable hearing loss, loss of vision and ovarian failure. 

The severity of the conditions was scored using the Common Terminology Criteria for Adverse Events (version 3). This is a scoring system developed through the National Cancer Institute by a multidisciplinary group and is widely used to score both acute and chronic health conditions in patients with cancer and survivors of all ages. (Cancer Therapy Evaluation Program, Common terminology criteria for adverse events, version 3.0.) The system grades conditions as mild (grade 1), moderate (grade 2), severe (grade 3), life-threatening or disabling (grade 4), or fatal (grade 5). For our study, a total of 137 health conditions were scored. 

2) Line 93: Please elaborate on previous programme that addressed psychosocial challenges and explain why wilderness therapy was chosen.

- We have now included additional information on existing psychosocially oriented programs and how a nature-based intervention under study fits into this history. Also, see the following section 1.3 where we justify in detail why we are testing a nature-based therapy.

Lines 99-105 “Due to a host of psychosocial stresses following cancer, psychosocial guidelines recommend providing psychosocial interventions to survivors.(29) A wide variety of interventions exist to address long term physical and emotional sequelae of cancer including educational(30) cognitive behavioral, (31) physical exercise(32-34), health behavior change(35) and social support programs. A number of nature-based programs exist, such as cancer camps and adventure therapy (usually for young adult cancer survivors) and are reviewed here.(36) Few programs explicitly evaluate the role of nature in healing. 

3) Line 147: Why money situation evaluated in the baseline?

- Psychosocial interventions have not always been accessible to all patients regardless of socioeconomic status (SES). This program made it a priority to provide all aspects of the program for free to remove SES barriers. 

4) Line 153: It is suggested that Table 2 be placed in the supplementary file rather than the main text.

- Good idea. It is now labeled S1 Appendix, Table 2. 

5) Line 292: It is recommended to present the contents of Table 3 in text.

-Since we put Table 2 in the S1 Appendix, the Recruitment and Tracking Table is now Table 2 (not Table 3). Typically, one does not put into text content that can be read from the table but in deference to the reviewer we now describe Table 2 results. Here is the edited text from Section 3.1, Lines 302-308. 

Lines 326-334 (new lines are underlined) “The SYATS recruitment goal was eight participants (the Forest Service permit maximum group limit is 12, which includes 8 participants and 4 staff members. Hospital staff (nurses, social workers, and ambassadors) approached teens/families and introduced the wilderness program. A total of 21 teenagers, aged 12-17 were referred to, or expressed interest in participating in SYATS. Among this group, the cancer diagnoses included: brain tumor, leukemia, germ cell tumor, and lymphoma among others. Among the 21 with interest in SYATS, 4 (19%) were ultimately not eligible, 6 (35%) did not contact SYATS again, 3 (18%) were not interested or had a scheduling conflict and 8 (47%) signed up (See Table 2). Retention was 100% with all eight teens participating from day 1-9.” 

6) Line 384: Table 5, please add the actual mean score along with range

- In (now) Table 4, we have added the mean scores and range of mean scores for baseline, day 9 FU, and 3-mo FU. We also still include the differences between baseline and the two follow-ups time-points.

7) In the discussion, suggest to compare the content and results of the current wilderness programme with previous programme.

- We now add the following information.

Lines 674-677. “The SYATS program provides preliminary data that is in accordance with data from other outdoor or adventure therapy programs including increased social involvement, increased self-esteem, self-confidence, self-efficacy, social support, and physical activity. (35)”

As described on page 29, previous programs are substantially different. They were not aimed at this age group, for this timespan, nor did they include an evaluation of the role of nature as well as mindful/reflective activities (journaling, meditation, yoga, meditation). None used the strongly validated PROMIS measures. See Scoping review referenced above (Jong, 2021).

Reviewers' comments:

Reviewer's Responses to Questions

Comments to the Author

1. Is the manuscript technically sound, and do the data support the conclusions?

Reviewer #1: Yes

Reviewer #2: Yes

Reviewer #3: Yes

Reviewer #4: Yes

Reviewer #5: Yes

2. Has the statistical analysis been performed appropriately and rigorously?

Reviewer #1: Yes

Reviewer #2: Yes

Reviewer #3: Yes

Reviewer #4: Yes

Reviewer #5: N/A

3. Have the authors made all data underlying the findings in their manuscript fully available?

Reviewer #1: Yes

Reviewer #2: No

Reviewer #3: Yes

Reviewer #4: Yes

Reviewer #5: No

In the resubmission we have described how data can be accessed after contacting the SYATS Medical director. 

4. Is the manuscript presented in an intelligible fashion and written in standard English?

Reviewer #1: Yes

Reviewer #2: Yes

Reviewer #3: Yes

Reviewer #4: Yes

Reviewer #5: Yes

5. Review Comments to the Author

Reviewer #1: 

1-The authors should explain about how to screen the qualilative data and should show the data that related or not related between participants.

2-In discussion part the authors should clearly describe why SYATS program that effect in participant.

- In the results, we have tried to exemplify the existing variety between (and sometimes within) individuals within quotes regarding feasibility, but also in terms of representation in themes and categories.

Reviewer #2: Overall, a strong piece of writing that might use some tweaking in terms of flow and presentation.

The authors may also provide research implications.

Overall, the writing is strong, but the flow and presentation could be improved.

As we have edited the article we worked to improve the flow and presentation by removing redundancy, editing syntax, and shortening. 

The authors may also provide research implications.

+Reviewer #3: Program evaluation of a wilderness experience for adolescents facing cancer: A time in nature to heal, connect and find strength. This manuscript good presentation, Overall, this paper has clear research issues, appropriate study methods, and presents interesting results.

Thank you!

+Reviewer #4: Program evaluation of a wilderness experience for adolescents facing cancer: A time in nature to heal, connect and find strength. Overall clear, concise and interest to reader. Its interesting for publish jouraal. Accept

Thank you.

Reviewer #5: The authors have provided a detailed description of a program of a wilderness experience for adolescents with cancer. the intent is noble and good- and the program is described discursively in a study of 8 participants, which really constitutes a preliminary/feasibility study. 

- We agree with the reviewer comment that the study is a preliminary/feasibility study and we hope we made that clear throughout where we mentioned this in the abstract, methods and discussion sections.

The small number of subjects is not really analyzed- nor evaluated in a systematic way beyond the success of participation. the manuscript describes the success of the project and provides ample- perhaps too many detailed quotations. the manuscript could benefit greatly from substantial shortening- and moving many of the anecdotal quotes to a series of supplementals documents organized by the questions asked- and the responses. as presented, it is repetitive and works against its purpose- a demonstration of the feasibility and utility of a program- that could be extended in scope.

- We have now moved material that might be redundant into the S1 Appendix Table 3 Under Section 3. We originally chose to organize the material around a list of criteria often used to assess pilot studies. (National Center for Complementary and Integrative Health. Pilot Studies: Common Uses and Misuses. Framework for Developing and Testing Mind and Body Interventions 2019 Washington, DC: NCCIH; 2019). We outlined these criteria at the end of the introduction (lines 130-135). We still think it makes sense to organize the material this way as it reflects that the evaluation criteria were thoroughly addressed. It also reflects the way we outlined the structure of the manuscript at the end of the introduction section where we say “The primary aim of the study is to describe the feasibility, acceptability, and safety of See You at the Summit (SYATS) wilderness program. The secondary aim describes the impact on participants including a quantitative assessment of global and mental health as well as qualitative findings and observations describing the program participation experience of SYATS participants. Starting on page 21, Line 472 we organized the qualitative analysis around themes that emerged which is a fairly standard way to report qualitative findings. The themes are described in Table 5. 

 the numbers are too small to establish any rigorous statistical analyses.

- We are in full agreement with the reviewer. The number of participants (while meeting our enrollment goal) was too small to draw any statistically significant conclusions about effectiveness. This is intended as a pilot study allowing the SYATS team to work out the details of the intervention delivery and evaluation components. Any statistical work will help us to understand trends and help us to calculate sample sizes for future studies, but cannot determine effectiveness. We hope we made that clear in a number of places in the manuscript (in the methods, results and discussion sections). I include that material below to show that the authors fully acknowledge the limitations of the small sample. 

Methods: Lines 151-153. We say “This data should be used for future sample size calculations and for descriptive purposes only given the small sample size.”

Results: Line 456. “While the small sample size precluded meaningful significance testing, PROMIS baseline data showed ….. As can be the case with small numbers, the mean changes can be strongly influenced by a few persons.”

Discussion, Line 701 “As expected, small numbers limit conclusions about the program impact,…” and Lines 770-776 “However, in this small group, we caution against over interpretation of mean PROMIS sub-scale scores since a few people can alter the group mean. Use of multimethod assessments provided rich data to better understand where participants stand in relation to norms (PROMIS) and to understand the lived experience.”

The final sentence of the abstract offers a vague statement that should be reworded. "controlled" studies is confusing and should not imply a randomization or standard clinical study for a handful of outcomes. Please be careful here and consider the next steps- is it continue the program or is it to develop clear and statistically compelling design- which would be very hard given the heterogeneity of patients, treatment schedules and timing of the structured experiences.

- Related to the final statement of the conclusion in the abstract—we have edited that statement to clarify our meaning. It now reads (Line 51) “Larger randomized controlled studies are needed to learn whether these programs can improve psychosocial outcomes.” We understand why you might say this and recognize the challenges in using the gold standard RCT study design in situations where it is difficult to blind participants and staff. That said, we do want to note that several co-authors have been involved in another similar wilderness intervention using a randomized controlled trial design that involves a wilderness adventure vs. a relaxing hotel spa retreat. Yes, you are correct, it is challenging, but this new trial was funded, and the intervention has been delivered with success. (Manuscript is in progress). The protocol reference is listed here. 

Jong MC, Mulder E, Kristoffersen AE, et al. Protocol of a mixed-method randomised controlled pilot study evaluating a wilderness programme for adolescent and young adult cancer survivors: the WAYA study. BMJ Open 2022;12:e061502. doi:10.1136/ bmjopen-2022-061502

6. PLOS authors have the option to publish the peer review history of their article (what does this mean?). If published, this will include your full peer review and any attached files.

Do you want your identity to be public for this peer review? For information about this choice, including consent withdrawal, please see our Privacy Policy.

Reviewer #1: No

Reviewer #2: No

Reviewer #3: No

Reviewer #4: No

Reviewer #5: No

---

## [Editor Report · Decision Letter 1]

7 Sep 2023

Program evaluation of a wilderness experience for adolescents facing cancer: A time in nature to heal, connect and find strength

PONE-D-22-28734R1

Dear Dr. Jong,

We’re pleased to inform you that your manuscript has been judged scientifically suitable for publication and will be formally accepted for publication once it meets all outstanding technical requirements.

Kind regards,

Cho Lee Wong, PhD

Academic Editor

PLOS ONE

Additional Editor Comments (optional):

Thank you very much for addressing comments from editor and reviewers.
---

## [Editor Report · Acceptance letter]

25 Sep 2023

PONE-D-22-28734R1 

Program evaluation of a wilderness experience for adolescents facing cancer: A time in nature to heal, connect and find strength 

Dear Dr. Jong:

I'm pleased to inform you that your manuscript has been deemed suitable for publication in PLOS ONE. Congratulations! Your manuscript is now with our production department. 

Kind regards, 

on behalf of

Dr. Cho Lee Wong 

Academic Editor

PLOS ONE